# Voltage Stability Control Based on Angular Indexes from Stationary Analysis

Gabriel J. Lopez *, Jorge W. González, Idi A. Isaac, Hugo A. Cardona and Oscar H. Vasco

Research Group on Transmission and Distribution of Electric Power, Universidad Pontificia Bolivariana, Medellin 050031, Colombia
* Correspondence: gabriel.lopez@upb.edu.co

**Abstract:** This paper presents a novel methodology for the calculation of angular indexes of an electrical system from stationary analysis, using load flow and nose curves (P–V) in each of the buses of the system to perform control actions and preserve or improve voltage stability. The control actions are proposed considering a novel method based on the concepts of the cutset angle (CA) and center of angle (COA). The target is a fast estimation of voltage-stability margins through an appropriate angular characterization of the whole system and for each load bus with a complete network and N-1 contingency criteria. The most significant enhancement is that the angular characterization is based on the COA, which is related to the angular dynamics of the system, and indirectly reflects the inertia and the respective angles of the generator rotor, as well as the impact on the angular equivalent-system model. Simulations showed that the COA is an important index to determine the location of occurrence of the events. The COA can also help aim where control actions, like the amount of load shedding, should be carried out to remedy the voltage problems. The proposed method is assessed and tested in the benchmark IEEE 39-bus system.

**Keywords:** cutset angle; center of angle; angular index; load flow; voltage stability; angular characterization; IEEE 39-bus system; PMU; synchrophasor; power-system stability

---





## 1. Introduction

In general, the stability of the power system has been classified from the causes that lead to instability [1,2]. However, any type of instability cannot be caused solely by an angular, voltage, or frequency problem. In highly stressed power systems, one form of instability can give rise to another, even generating cascading events that result in power-system collapses [3].

Although the classification of the stability of the electrical system is effective [1], and the means to face the complexities of the problem are convenient, the global stability of the system must always be the basis for the solution of any category of stability challenges (angle, voltage, or frequency), but not at the expense of affecting another stability category. It is essential to look at all aspects of the stability phenomenon from more than one point of view, inclusively, integrating the bus voltage angles more effectively.

In this article, a methodology is proposed to carry out stability analyses with the criterion of maintaining a safe voltage by applying classical methods of long-term voltage stability [1,2]. By using angular indexes calculated from synchrophasor measurements [3–7], this study aims to develop a wide-area control system as other references have initiated [5,8].

In many cases, static analysis can be used to estimate stability margins, which can be done even more so using Thevenin equivalents alongside loadability curves or control methods [9,10], identify factors that influence stability, and analyze a wide range of system conditions and a large number of scenarios [5,11–13]. When the time of the control actions is important, the analyses must be complemented by quasi-static, dynamic, or transient simulations [14,15].

The contributions of this paper are as follows:

We propose a novel method based on the cutset angle (CA) and center of angle (COA) [16–19] in the field of voltage stability for real-time operations, including bus voltage magnitudes and phase angles as obtained from PMUs. The system framework is simple to implement.

The study proposes a global voltage-stability margin to be estimated for the whole system and for each load bus with a complete network and N-1 contingency criteria. It was observed that the calculation of the indexes, especially the COA, yields the characterization of an electrical power system. It is even possible to determine if a contingency is critical once a detailed prior characterization of the system is available. In addition, the COA proved to be an important index in determining the location of occurrence of events, and where control actions should be carried out to alleviate the voltage problem in this case.

Compared with previous work [20], the most significant enhancement is that our angular characterization is based on the COA, which is related to the angular dynamics of the system, and indirectly reflects the inertia and the respective angles of the generator rotor, as well as the impact on the angular equivalent-system model [21,22]; this helps to carry out control actions based on load shedding and COA characterization to avoid voltage instabilities. This method can be applied to systems with a high penetration of renewables because a wide variety of controls that emulate inertia are currently being proposed [23–26].

The proposed methodology is promissory for the multiplicity of wide-area applications. Reference [20] does not include equivalent angles that allow for the characterization of the system to take control actions and keep voltage stability; the general proposal in [20] is to evaluate angle severity and thresholds that are obtained by considering the bulk transfer of power throughout the area, as limited by overloads of lines inside the area [20].

The proposed method here is assessed in the benchmark IEEE 39-bus system. The simulation results demonstrate an accurate evaluation with a significantly reduced system-response time.

Section 2 introduces some basic concepts of voltage-stability theory, P–V curves, recommended actions to improve voltage profiles in an electrical system, and the calculation of angular indexes. Sections 3 and 4 present a methodology and its application for the calculation of indexes with stationary analysis. Section 5 proposes a methodology for applying control actions based on angular indexes, Section 6 shows the simulations, and Section 7 shows the results obtained using the proposed methodology. The last section includes the conclusions.

## 2. Voltage Stability, Controls, and Indexes

Voltage stability can be classified as a short-term or long-term phenomenon. The study-time horizon for this type of problem can vary from a few seconds to tens of minutes [1].

Long-term voltage stability involves slow-acting equipment, such as transformers with load tap changers (LTCs), thermostatically controlled loads, and generator current limiters (OXLs or Overexcitation Limiters). The study time can extend to several or many minutes, and long-term simulations are required for the analysis of the dynamic performance of the system.

### 2.1. Power-System Voltage Stability

It should be noted that the action time varies for different control measures. In general, in order to achieve a long-term balance in voltage, the load can be restored to a constant power type by the voltage controlling action of LTCs. The amount of power recovery, which in turn satisfies an allowable voltage band, depends on the voltage setpoint on the LTC. Sometimes, the load amount will even have to be reduced.

The pre- and post-contingency PV curves are presented in Figure 1 with solid lines. The pre-contingency operating point is A and the demand corresponding to that point is PA. Long-term (constant power type) curves or characteristics are shown as vertical dashed

lines and the short-term equivalent are shown as quadratic dashed lines. P–V curves play a prominent role in understanding and explaining voltage instability; it could be said that they are the most used worldwide for these analyses [4,12,14,27–29].

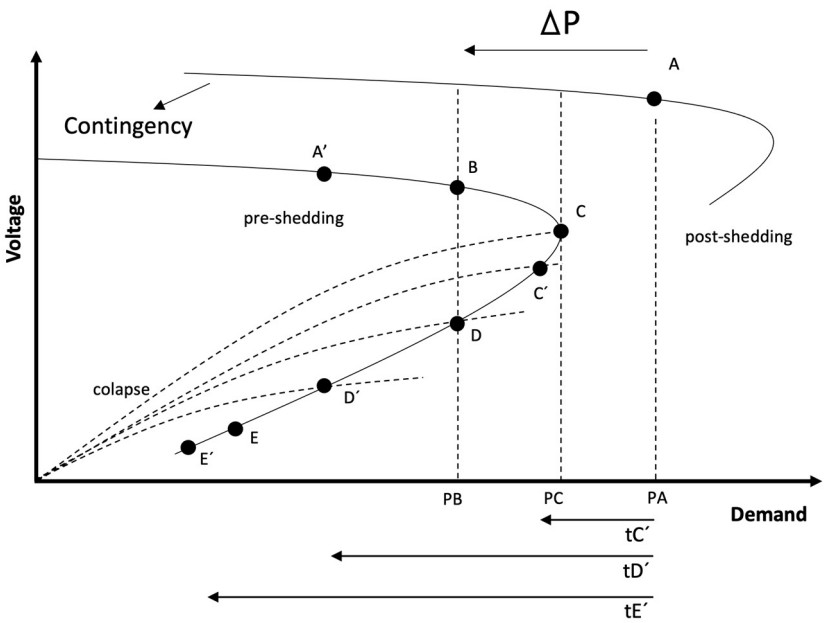

**Figure 1.** Load shedding effect after a contingency [4].

In the event of a contingency, there will be long-term instability at operating point A, because the demand PA is greater than the maximum power PC. By varying the LTC setpoint and shedding a load amount, it is possible to consolidate the new operating point B.

Load shedding is a case similar to load reduction through LTC with an extra complexity, as load is shed to improve the long-term characteristic, while the short-term characteristic also changes. This allows a little more time to perform control actions. The analysis is limited to the goal of restoring stable equilibrium, but the results also apply to the goal of stopping system degradation.

The load-shedding effect can be seen in Figure 1, where a restoration is assumed, by the action of the power system to the type of constant power load. When the amount of load shed is the minimum, PA–PC, the time limit for the restoration of an equilibrium is tC′, which is the time required for the short-term characteristic to reach point C′, so that the short-term load characteristic after shedding moves through point C. After time tC′, the value of the load to be shed increases. This is also true for the difference between the short-term characteristics for pre- and post-shedding. Therefore, load is shed at PA–PB, the time limit to restore stability at B is tD′ where point D′ corresponds to the short-term pre-shedding characteristic, so that the post-shedding feature moves toward point D.

Finally, it has to be noticed that the time tE, which is the last limit for control actions, remains unchanged because at E, the system is unstable.

Short-term voltage instability lasts a few seconds, whereas long-term voltage instability can last some minutes. Both are generally considered to be relatively quick for control actions to be undertaken. However, the time it takes for long-term instability, while it is short for an operator action, is long enough for an efficient code to run and scan the problem, warn the operator, and propose or apply corrective actions.

### 2.2. Angular Indexes

The cutset angle and the central angle are special angular indexes proposed and discussed in [16–19,30–32].

Cutset angles/areas conjugate plenty of angular measurements in a grid. This method yields weighted angles and the susceptance among two areas [16]. A Krön reduction has to be applied [33] to yield an effective relation between two areas as a function of the power flows and transmission line susceptance/angles in the boundary nodes of the system.

Figure 2 shows a basic diagram in which the equations of the cutset angle are explained.

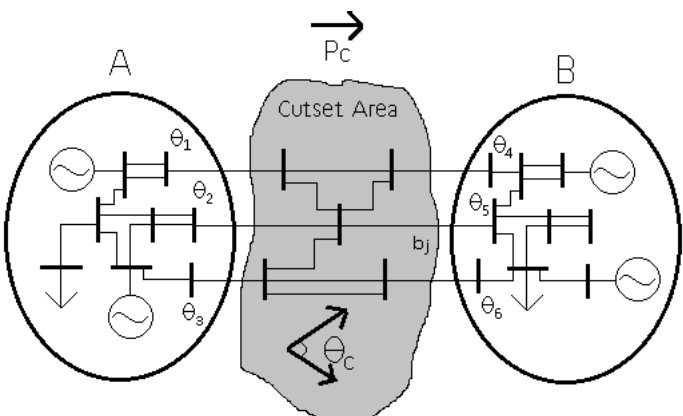

**Figure 2.** Cutset angle between two areas.

Figure 2 shows two areas named A and B, divided by lines denominated "cutset lines" in the cutset area [16] This cutset area is spread out all over the network. Equations (1)–(3) are obtained by dc load flow [17]. An AC load-flow solution was used for this study.

$$b_c = \sum_{j \in C} b_j, \tag{1}$$

$$\hat{\theta}_c = \sum_{j \in C} \frac{b_j}{b_c} \hat{\theta}_j, \tag{2}$$

$$P_c = b_c \hat{\theta}_c, \tag{3}$$

The COA algorithm is based on the calculation of angular differences between busbars of a network, considering a reference angle [16,17,34]. A main objective is to determine transient angular instability. This analysis is related to angular dynamics, which reflects the inertia and the angles of generator rotors, as well as the influence on the angular equivalent-system model. The COA can be interpreted as the center of mass of a body, being a universal reference, with dynamic phenomena. Disturbance could lead the system far from the center of mass, destabilizing the system.

For the proper use of COA concepts, reduction methods are required to relate several areas of a power system based on power flows, transmission line susceptance, and voltage angles in the boundary busbars, as shown in Figure 2.

As mentioned before, the COA is a real-time reference. The variation from the center of inertia of the phase angle can be calculated using the COA as defined in (4) and (5) [35].

$$\delta_{COA} = \frac{\sum_{i=1}^{N} \delta_i H_i}{\sum_{i=1}^{N} H_i}, \tag{4}$$

$$\delta_{COA} = \frac{\sum_{i=1}^{N} \delta_i P_i}{\sum_{i=1}^{N} P_i}, \tag{5}$$

where in the case of generator *i*:
$\delta_i$ is the rotor angle
$H_i$ is the inertia time constant
$N$ is the total number of generators in the areas of the system

$\delta_{COA}$ is the central angle (or "Center of Angle").

As (4) could be complicated for a real-time calculation, it may be useful, for each plant, to estimate the generation units' dispatching power. It should be performed as a function of time to obtain an equivalent H in the areas of the network.

The angle in (4) can be replaced by power injection measurements due to the direct relationship of weighting factor H to the real power injected [6]. As a consequence of the difficulty in measuring the internal angle of the rotor in real time, it is supposed that it has the same tendency as the voltage angle measured in the busbar (if the rotor angle increases or decreases, then the angle bus also increases or decreases, respectively).

It is possible to obtain the angular difference between the rotor and busbar using the internal voltage behind the machine synchronous reactance and produce a replica of the machine angular displacement. Phasor diagrams in Figure 3 provide the latter explanation. The COA calculation is therefore expressed in terms of variables measurable by PMUs.

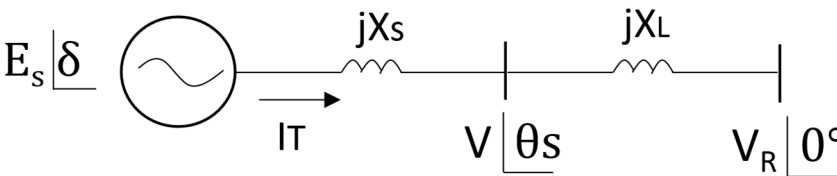

**Figure 3.** Equivalent system diagram (generator and infinite bus).

This system diagram model neglects saliency effects and stator resistance. Therefore, if the saliency is neglected [36]:

$$X_S = X_d = X_q, \tag{6}$$

A voltage law equation must be written around each of the meshes, as follows (see Figure 4):

$$E_s \lfloor \delta = I_T j(X_S + X_L) + V_R \lfloor 0°, \tag{7}$$

$$V \lfloor \theta_s = I_T j(X_L) + V_R \lfloor 0°, \tag{8}$$

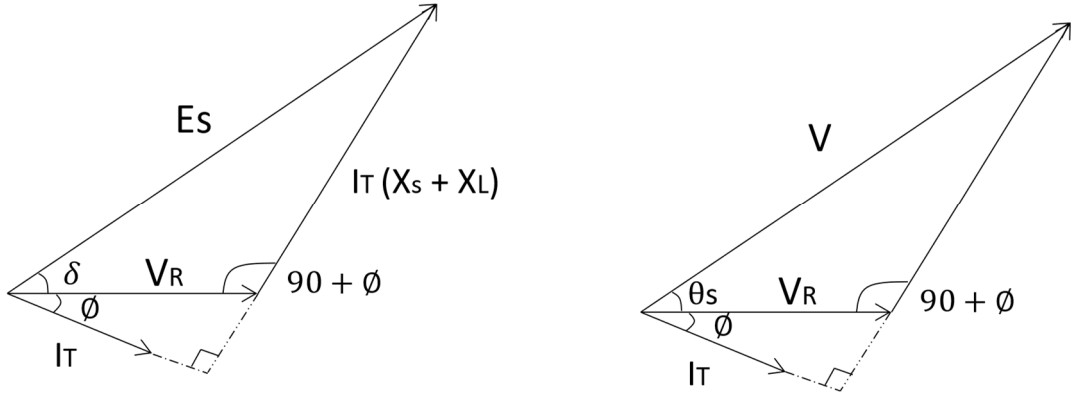

**Figure 4.** Steady-state phasor diagrams for Equations (7) (**left**) and (8) (**right**).

Applying the law of sines and cosines, it is possible to find the relationship between $\delta$ and $\theta_g$ (9).

$$\sin \delta = \frac{|V|}{|E_s|} \frac{(X_S + X_L)}{X_L} \sin \theta_s, \tag{9}$$

where
$X_S$ is the synchronous reactance
$X_L$ is the transmission line reactance
$|V|$ is the magnitude of the voltage angle bus
$|E_s|$ is the magnitude of the internal voltage of the synchronous machine

$\delta$ is the internal rotor angle

$\theta_s$ angle bus is measured by PMUs:

where

$$\frac{|V|}{|E_s|} \frac{(X_S + X_L)}{X_L} = K_g, \tag{10}$$

Hence,

$$\sin \delta = k_g \sin \theta_s, \tag{11}$$

And Equation (11) is applicable for small angles:

$$\delta = k_g \theta_s, \tag{12}$$

Finally, Equation (12) clearly shows the relationship between the rotor angle and the bus angle [37]. The tendency of the angle at the terminal machine is proportional to the rotor angle; however, it depends on the relationship between the voltages and reactance of the system (kg factor). Figure 5 shows the tendency of two load conditions.

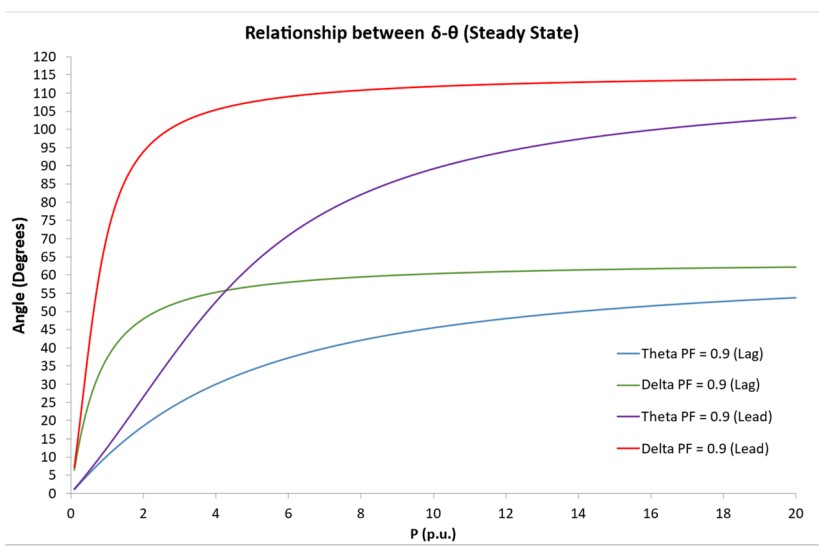

**Figure 5.** Relationship between $\delta$ and $\theta$ for different load conditions.

For any specific area i of the power system, the center angle is defined in Equation (13). Where $\delta^i_{COA}$ is the central angle for the area $i$, $\delta^i_j$ is the rotor angle of each generator, $P^i_j$ the power injected by the generator, and $N^i$ is the total number of generators in the area $i$.

The global center of inertia can be calculated solving (5).

$$\delta_C = \frac{\sum_{i=1}^{N} \delta^i_{COA} P^i}{\sum_{i=1}^{N} P^i}, \tag{13}$$

where N corresponds to all the areas and $P^i$ belongs to the total power generated in area i.

According to the COA of the system, a heuristic criterion to detect transient instability in real time is addressed. In case that the COA of an area is increased beyond a predetermined value from the center of inertia, it can be interpreted that the area has been separated from the rest of the system. An appropriate corrective action could be to either shed generation or load in this area [18].

It is advisable to develop a detailed characterization of the system in order to appropriate define thresholds for each area of the operation. In the same manner, it is important to clarify that the proposed method is compatible with any network in which inertia is obtained, either from synchronous generator data or from inertia constants emulated by

generation integrated by power inverters. Inertia emulation is a feasible function which is even required by some national operators nowadays [23].

## 3. Angular Characterization Methodology with Stationary Analysis

An angular characterization methodology is proposed from load flows (steady state) as mechanisms for taking control actions in the system.

It is proposed to generate safe operation thresholds (V $\geq$ 0.9 p.u.), considering contingencies N-1 for each of the proposed indexes, the central angle (COA) and the cutset angle (AngC).

The proposed methodology has the following stages:

### 3.1. Stage 1

The load flows and the voltage and load characteristics in each busbar of the electrical power system (P–V or nose curves) are calculated. These curves are obtained for different scenarios and for all possible N-1 contingencies. After that, the known angular indexes are calculated: the cutset angle and the central angle.

### 3.2. Stage 2

Demand scenarios are established at each intersection of the P–V curve with the safe voltage limit, see Figure 6.

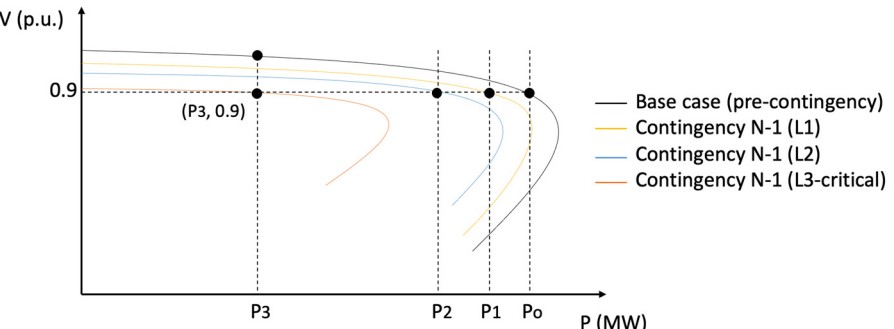

**Figure 6.** Determination of demand scenarios for the voltage limit (0.9 p.u.).

The worst contingency is sought, which is determined by the curve that reaches the lowest demand value: it is assumed that this is the one that exhibits the greatest stress for that particular condition because it represents the scenario that has the least possibility to stress the system in a greater proportion (close to the collapse point). Once the worst contingency has been determined, a vertical line is drawn that intersects the curve at 0.9 p.u., and the demand point (P3 in Figure 6) that ensures the operation limit is found.

Subsequently, using a relationship [P-theta (angular index)], the angles that maintain the voltage within the proposed limits (0.9 p.u.) are calculated, both for the curve of the base scenario and for the scenario of the worst contingency. The intersection of curves with point P3 in Figure 7 can be observed. Subsequently, the delta of the COA index is found. The above procedure is performed for different operating scenarios.

With these angular values, it is suggested to build maximum and minimum limit curves for the operating areas and for each scenario, in which voltages above 0.9 p.u. are ensured and the deltas of the COA are related, see Figure 8. In this way, it is ensured that any contingency N-1 is within the band limited by the maximum and minimum angular values for a specific load demand.

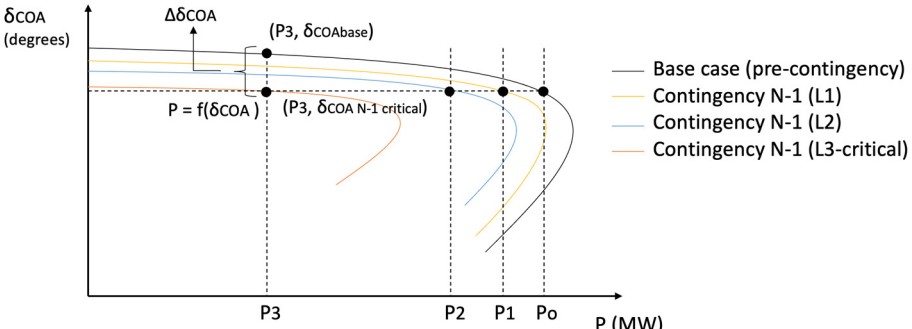

**Figure 7.** Determination of the angular index for different demand scenarios with voltage limit (0.9 p.u.).

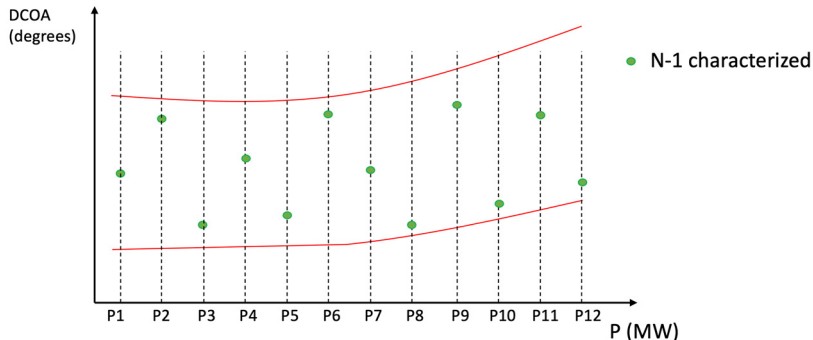

**Figure 8.** Determination of angular thresholds for a demand scenario with contingencies N-1.

*3.3. Stage 3*

This stage is for angular data collection, with which it is sought to carry out off-line training that allows for a complete characterization of the electrical power system, regardless of its operating or topological conditions.

## 4. Application of Angular Characterization Methodology to the IEEE 39-Bus System

*4.1. IEEE 39-Bus System*

The proposed methodology is validated in the IEEE 39-bus System [6]. This system is composed of 10 generators, 19 loads, and 34 lines. The system parameters are taken from the book *Energy Function Analysis for Power System Stability* [36]. In a stable-steady and pre-contingency state with all its elements in normal operation, the system presents a generation of 6140.81 MW and a demand of 6097.10 MW.

This system can be divided into four operational areas in order to perform COA and AngC analyses, as shown in Figure 9.

Table 1 shows the data in normal operation and the pre-contingency for each operative area.

**Table 1.** Data of the operational areas in IEEE 39-bus system.

| Area | Busbars | Generation [MW] | Load [MW] |
|---|---|---|---|
| Area 1 | 1, 2, 3, 18, 25, 30, 37 | 790 | 704 |
| Area 2 | 17, 26, 27, 28, 29, 38 | 830 | 909.5 |
| Area 3 | 4, 5, 6, 7, 8, 9, 10, 11, 12, 13, 31, 32, 39 | 2170.8 | 2376.5 |
| Area 4 | 14, 15, 16, 19, 20, 21, 22, 23, 24, 33, 34, 35, 36 | 2350 | 2107 |

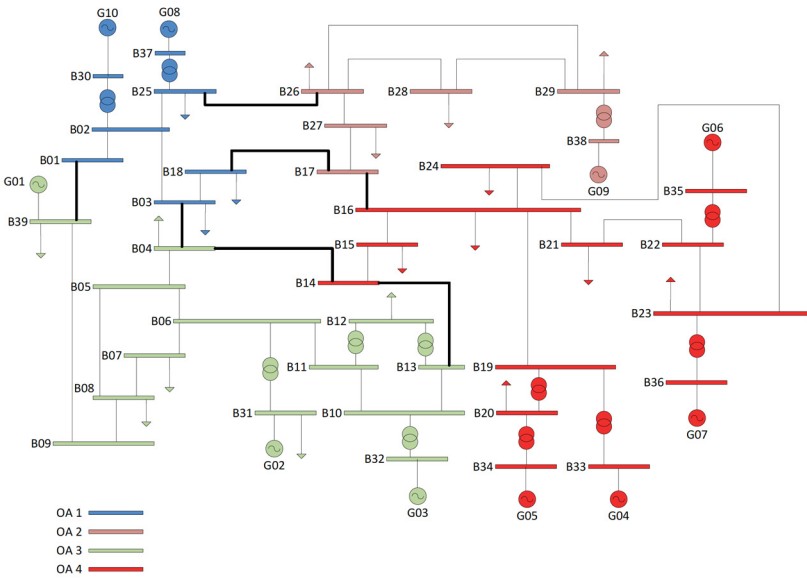

**Figure 9.** IEEE 39-Bus System.

*4.2. Angular Characterization in IEEE 39-Bus System*

With the help of DIgSILENT PowerFactory software, the P–V curves are calculated for the IEEE 39-bus System, and the indexes are calculated with different scenarios of demand increase in a certain area of the system (area 1, area 2 and area 3, the entire system, etc.). Reactive power limits of generators were considered for P–V curve tracing. This same process is carried out for contingency scenarios N-1; that is, scenarios of increased demand and the unavailability of one element of the system at a time.

It is important to mention that this paper presents the characterization for the central angle COA as little variability was observed for the cutset angle AngC.

Subsequently, for each scenario, the contingency that leads the voltage in any bus of the system to a lower demand is identified (graphically, it would be the P–V curve that first crosses the value of 0.9, see Figure 6). Once the event has been identified, the values of the indexes are captured, in this case the delta of the COA (DCOA, see Figure 7), and the demand and the angular value are obtained, up to which it is ensured that no contingency N-1 causes sub-voltages in the system (see Table 2).

**Table 2.** Parameters of the operational areas in IEEE 39-bus system.

| Area | Load [MW] | DCOA1 [°] | DCOA3 [°] | DCOA4 [°] | DCOA3 [°] |
|------|-----------|-----------|-----------|-----------|-----------|
| 1 | 1048.19 | −3.70 | 3.82 | −2.38 | 2.44 |
| 2 | 1008.86 | −1.63 | 4.60 | −4.70 | 3.46 |
| 3 | 2606.78 | −1.86 | 6.11 | −4.88 | 3.47 |
| 4 | 2157.88 | −1.38 | 6.37 | −5.43 | 3.36 |

Finally, the maximum and minimum values of the DCOA indexes are calculated, thus obtaining two final curves depending on the demand for each index, which represent the maximum and minimum values of the DCOA that can be given for different demands of each area of the system (a safe operating band, see Figure 8), which finally represents the operating ranges between which each index would move for the different contingencies N-1 of the system.

In Figures 10–13, the characterization carried out according to the described procedure is shown; that is, the variation for areas 1, 2, 3, and 4 of the DCOA central angle delta index.

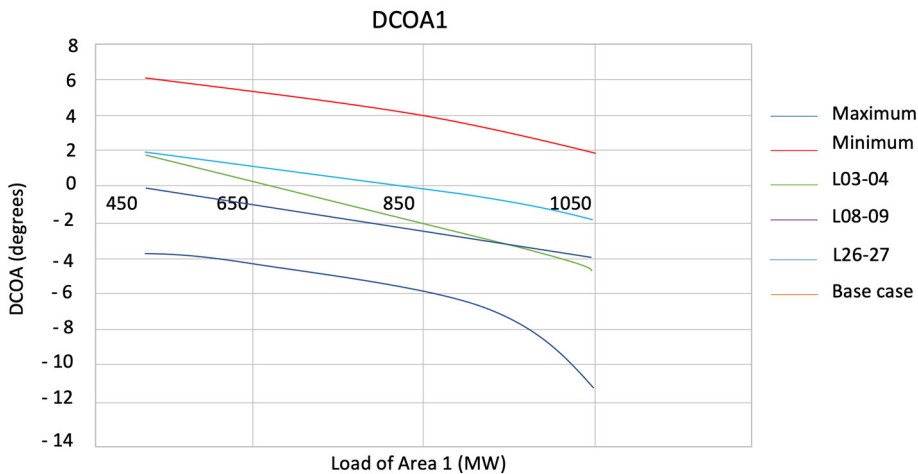

**Figure 10.** Angular characterization for contingencies N-1 in operational area 1.

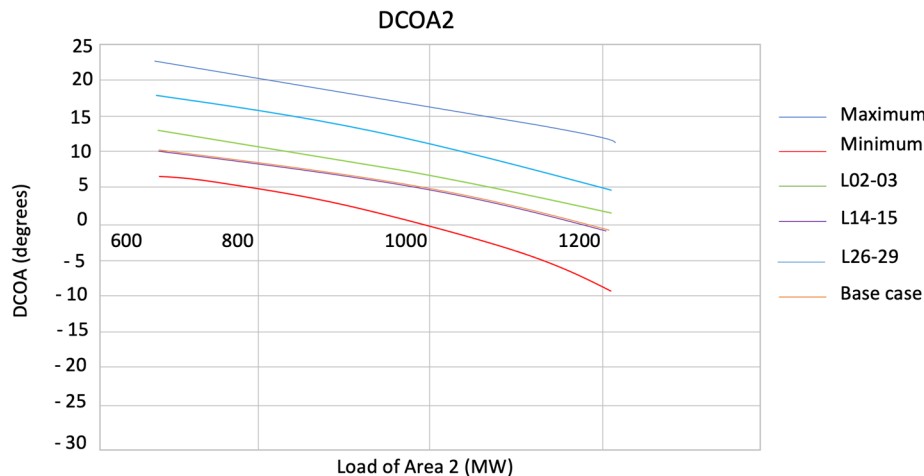

**Figure 11.** Angular characterization for contingencies N-1 in operational area 2.

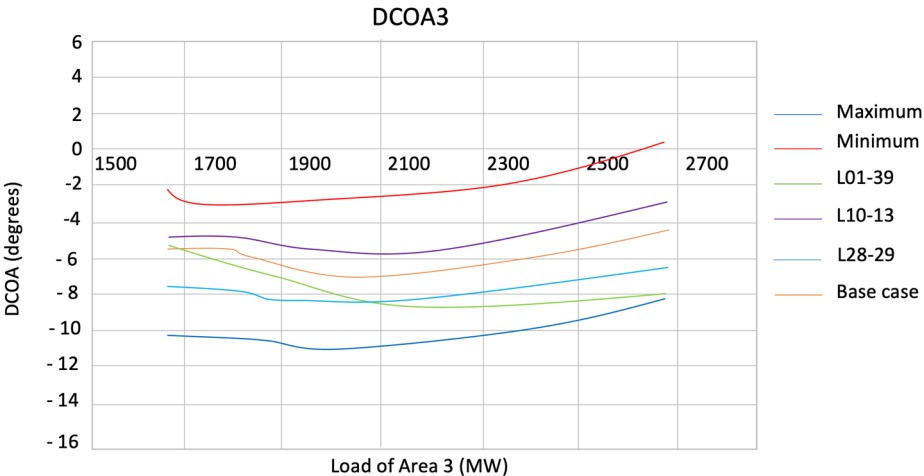

**Figure 12.** Angular characterization for contingencies N-1 in operational area 3.

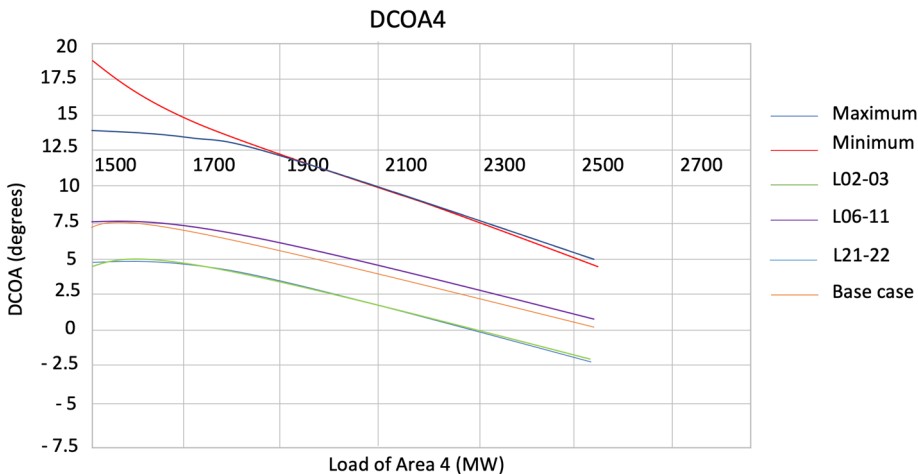

**Figure 13.** Angular characterization for contingencies N-1 in operational area 4.

## 5. Angular Characterization Methodology with Stationary Analysis

To evaluate the possibility of taking control actions through a stationary analysis, a methodology composed of the following stages is proposed:

### 5.1. Stage 1

This stage intends to carry out a stationary and dynamic analysis of the performance of the indexes for contingencies that were considered critical in the previous angular characterization. Where contingencies such as outages of transmission lines, generators, transformers, and busbars were carried out, the values of the indexes are taken in a stable and dynamic state in order to observe their behavior.

### 5.2. Stage 2

Once these critical contingencies are found, a load-flow analysis is carried out, and the P–V "nose curve" is plotted for each of them. The load flow shows a non-convergence for the evaluated scenario (because it is a critical contingency scenario). At this point, a dynamic simulation is performed, analyzing the behavior of the voltages and frequencies in the system, and its criticality is verified. It should be clarified that the behavior of the frequency is outside the scope of this methodology because only the voltage stability is being studied. However, it has been verified that, by taking the optimized load shedding and additional generation rejection actions, it is possible to stabilize the behavior of the frequency.

### 5.3. Stage 3

Subsequently, a load-shedding scheme based on the stationary load-flow analysis is carried out. The methodology to identify the place and percentage of the load to shed is based on the angular characterization, as explained above.

Once the criticality of the event is validated, the most affected operational area is identified by calculating the delta of the COA. This identification is made with the calculation of the deltas that are located outside the limit bands found in the characterization carried out.

In the next stage, the P–V "nose curves" are generated for all of the busbars of the affected area, and the weakest bars are identified due to the contingency performed (the weakest bar is identified with the P–V "nose curve" that reaches a bifurcation or a lower value of voltage). The load-shedding actions in this case are taken as shown in Figure 14, where:

a:  Point of maximum power in bus i before non-convergence for the base case.
a':  Safe operating point for the base case.
a":  Safe operating point for the base case. It is the initial condition.
b:  Minimum voltage point found for bus i after carrying out the critical contingency from point a".
b':  Safe operating point for critical contingency.

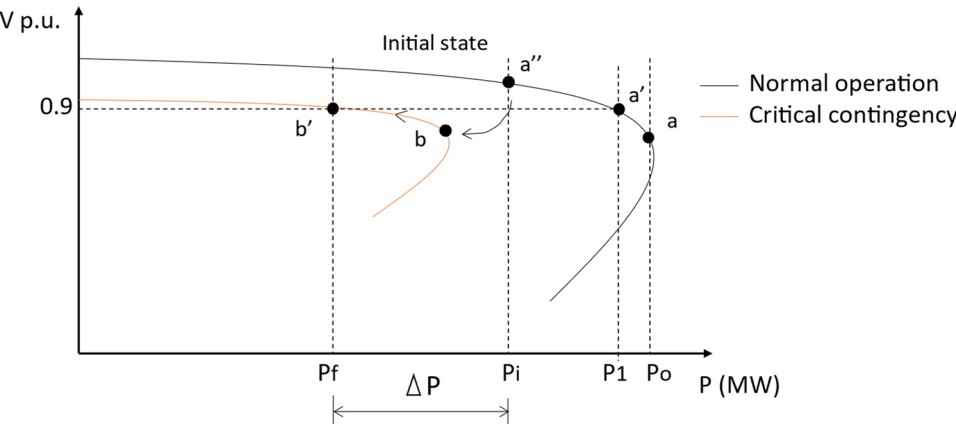

**Figure 14.** P–V curve for normal and contingency operation (determination of load shedding).

The load-shedding action is performed by finding the power delta from point a″ to point b′, and the percentage of load to be shed is calculated using Equation (14):

$$\%\text{load shedding OA}_i = \frac{\Delta P}{P_{area\_i}},$$ (14)

This percentage of load shedding is carried out for all loads in the affected area.

## 6. Simulation

The outage of transmission lines 5–8 and 6–7 was simulated, separating bus 7, 8, and 9 from the rest of the system, connected only by line 9–39, which caused a voltage drop in the busbars and the loads connected to them: 230 MW (bus 7) and 522 MW (bus 8).

The most affected area was identified according to the angular indexes (see Figure 17). It can be seen in Figure 17 that the contingency of lines 5–8 and 6–7 was outside the thresholds defined in the angular characterization for contingencies N-1 in operational area 3 (DCOA3). For the other areas, this event did not represent any problem in the bus voltages of the system (see Figures 15–18).

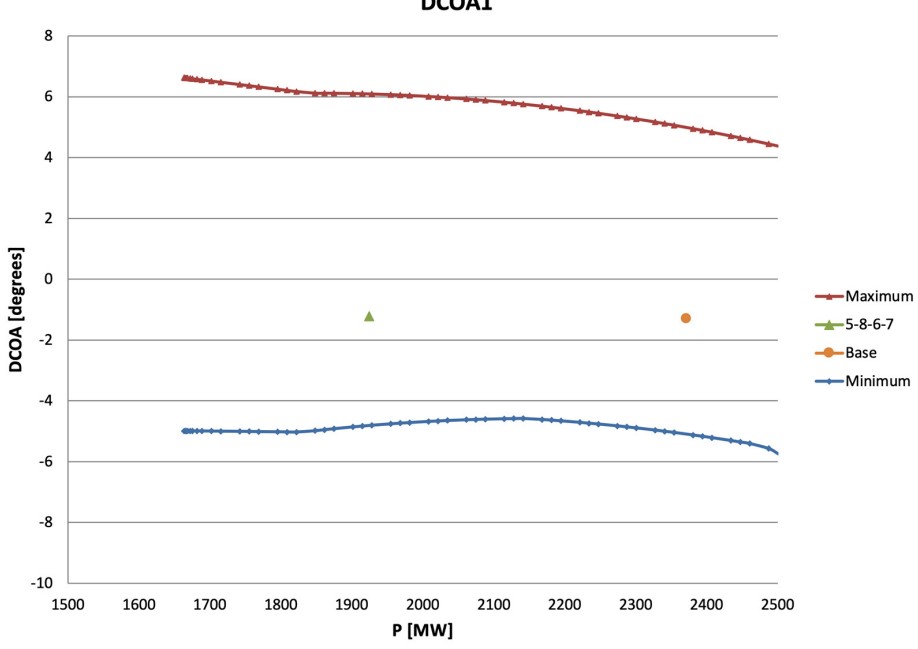

**Figure 15.** Angular threshold characterization for contingencies N-1 in operational area 1 (outages of lines 5–8 and 6–7).

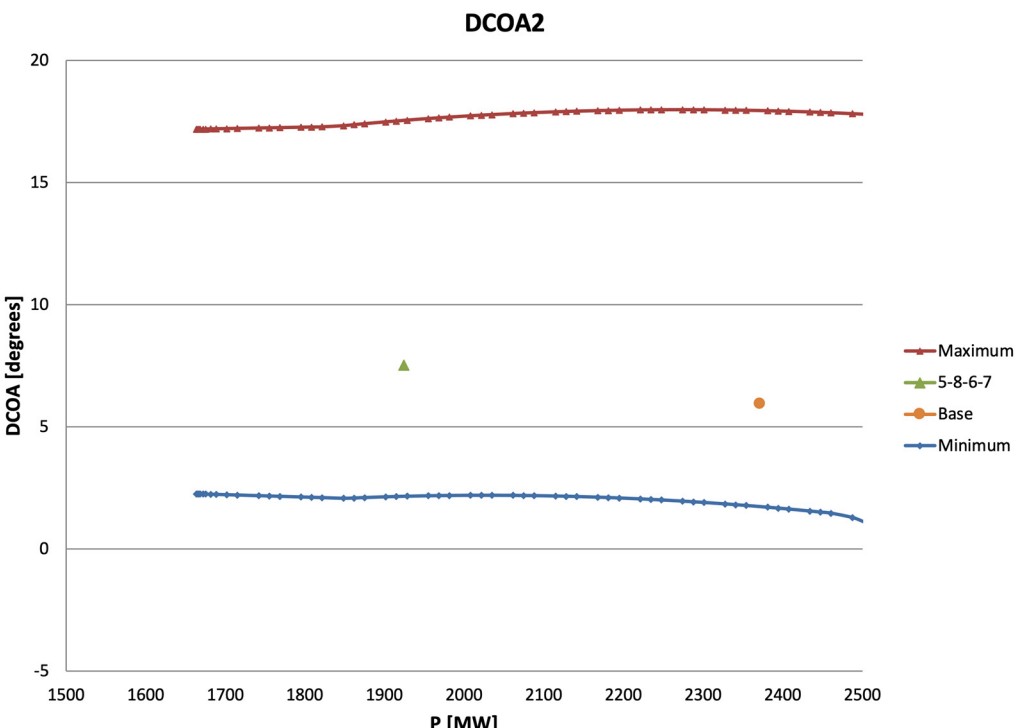

**Figure 16.** Angular thresholds characterization for contingencies N-1 in operational area 2 (outages of lines 5–8 and 6–7).

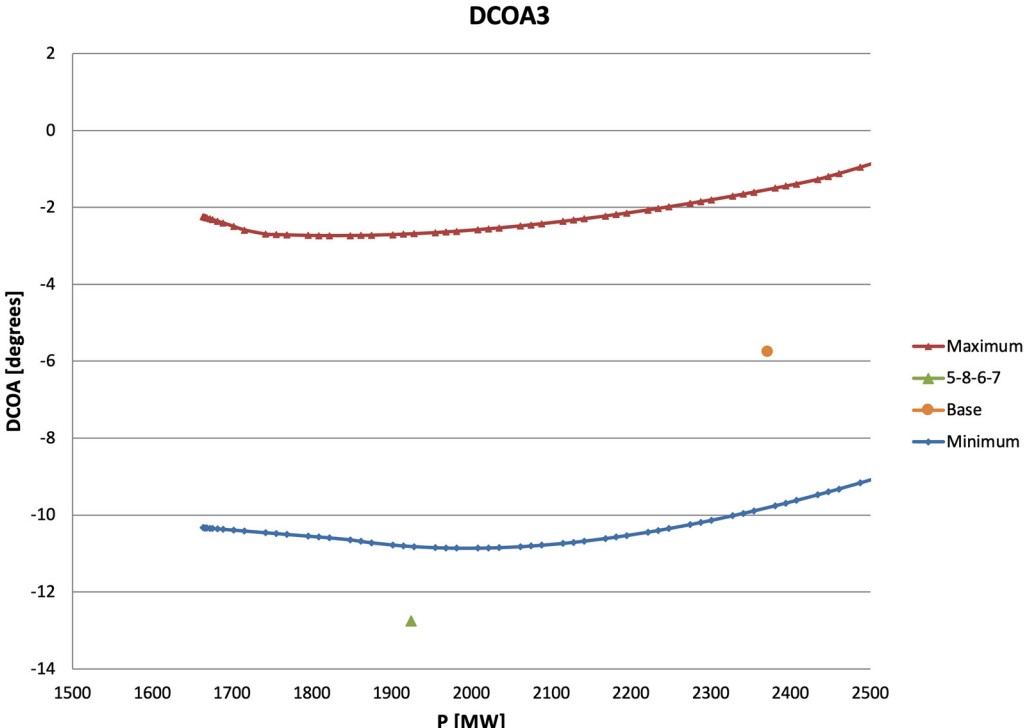

**Figure 17.** Angular thresholds characterization for contingencies N-1 in operational area 3 (outages of lines 5–8 and 6–7).

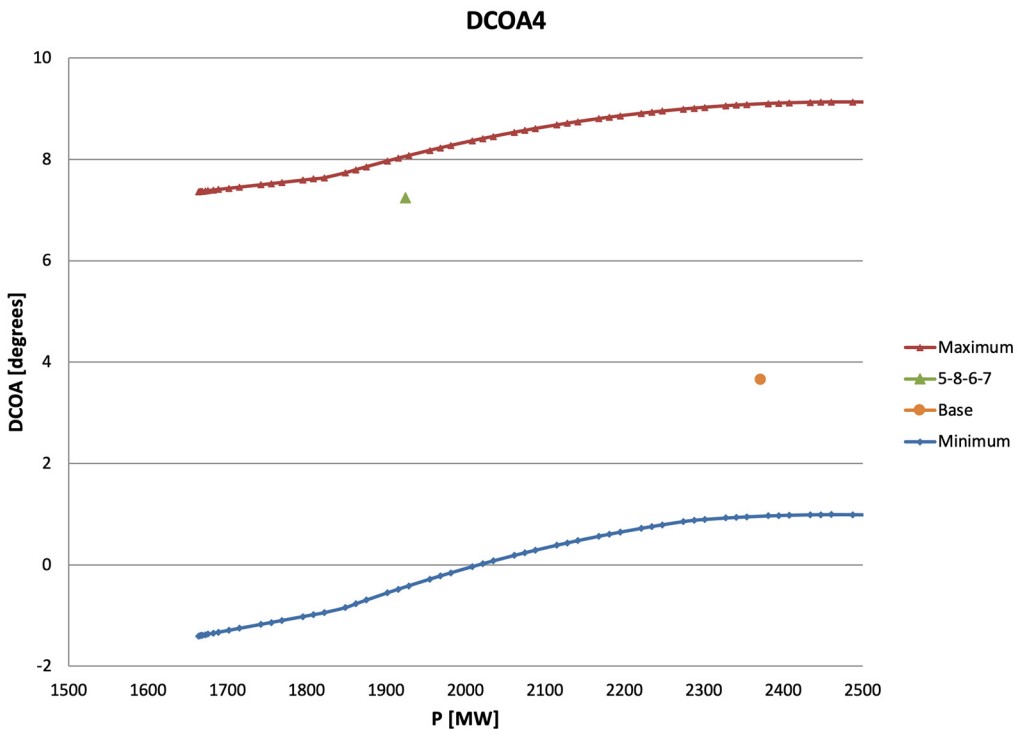

**Figure 18.** Angular thresholds characterization for contingencies N-1 in operational area 4 (outages of lines 5–8 and 6–7).

After identifying the most affected operational area (area 3), the amount of load to be shed was found.

From Figure 19, the control action was performed based on the ΔP and the total load of operational area 3:

$$\Delta P \approx 1000 \text{ MW}, \tag{15}$$

$$\text{P area } 3 = 2400 \text{ MW}, \tag{16}$$

$$\% \text{ Load shedding OA3} = (1000 \text{ MW})/(2400 \text{ MW}) = 42\%, \tag{17}$$

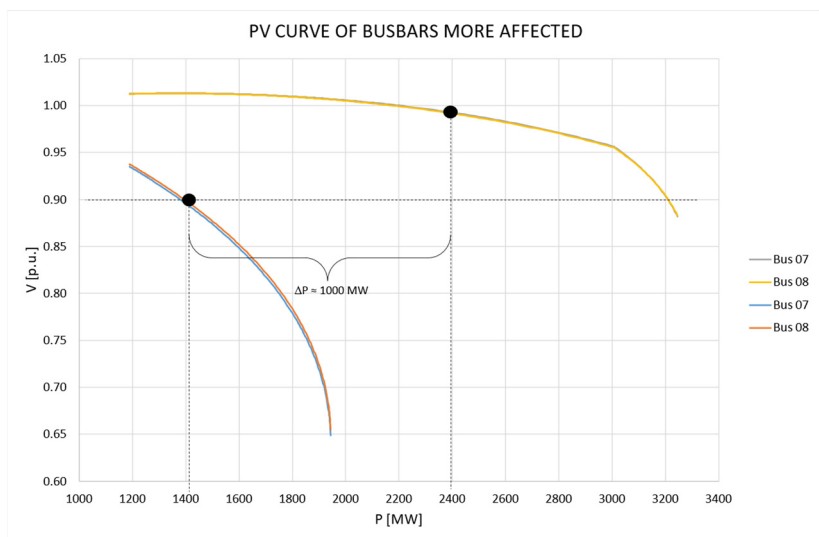

**Figure 19.** P–V curves for normal and contingency operation in operational area 3 (load shedding calculation).

## 7. Results

This section shows the behavior of the voltages in all busbars for different OAs of the IEEE 39-bus system. The results were carried out taking into account the angular characterization using the COA and DCOA for each load bus and the N-1 contingency criteria. In this case, the control actions were conducted using the classical P–V curves and their relationship with the COA and DCOA indexes. Then, the simulations results and figures were addressed and explained in detail.

Figure 20 shows the voltage results. It can be seen that, when performing the simulation with respect to time without control actions, there was a clear loss of stability, with oscillations and voltages below 0.9 in almost all the busbars of the system by operational area; voltages around 0.7 p.u. were particularly appreciated in area 3. When performing control actions in operational area 3 (OA3), a notable improvement in the voltage profiles was observed, damping the oscillation that appears in the system.

Comparative results are presented for our method against the classical literature P–V curve construction as it is a conventional methodology frequently used in the industry (see Table 3). It can be seen that the COA method is promissory for voltage-stability requirements in a power system.

**Table 3.** Voltage deviation comparison for the classical literature method and proposed COA method. Outages of transmission lines 5–8 and 6–7.

| Most Affected Voltage Bus in OA | Maximum Voltage Deviation w/o COA Method | Maximum Voltage Deviation with COA Method | Minimum Voltage Deviation w/o COA Method | Minimum Voltage Deviation with COA Method | Voltage Stabilization Time w/o COA Method [s] | Voltage Stabilization Time w/o COA Method [s] |
|---|---|---|---|---|---|---|
| 1 | 1.150 | 1.098 | 0.787 | 0.951 | 17 | - |
| 2 | 1.157 | 1.096 | 0.883 | 0.992 | 16 | - |
| 3 | 1.111 | 1.092 | 0.686 | 0.897 | >20 | 4 |
| 4 | 1.130 | 1.095 | 0.911 | 0.971 | 16 | - |

Without control actions　　　　　　　　　With control actions

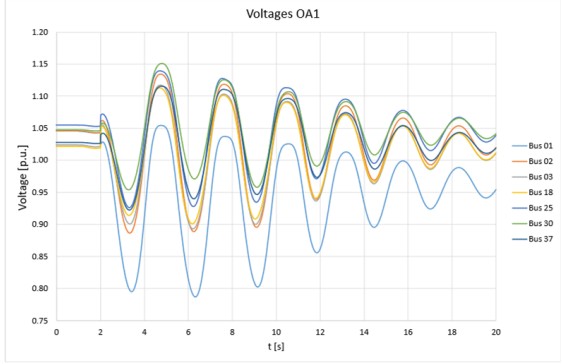 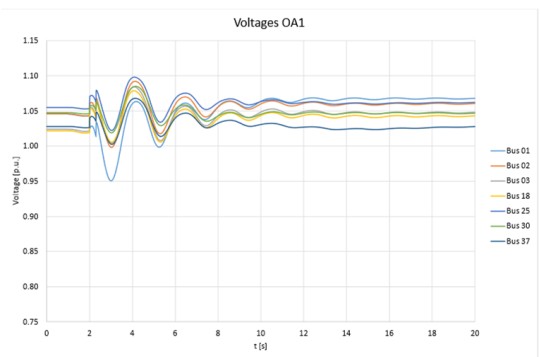

(**a**)

**Figure 20.** *Cont*.

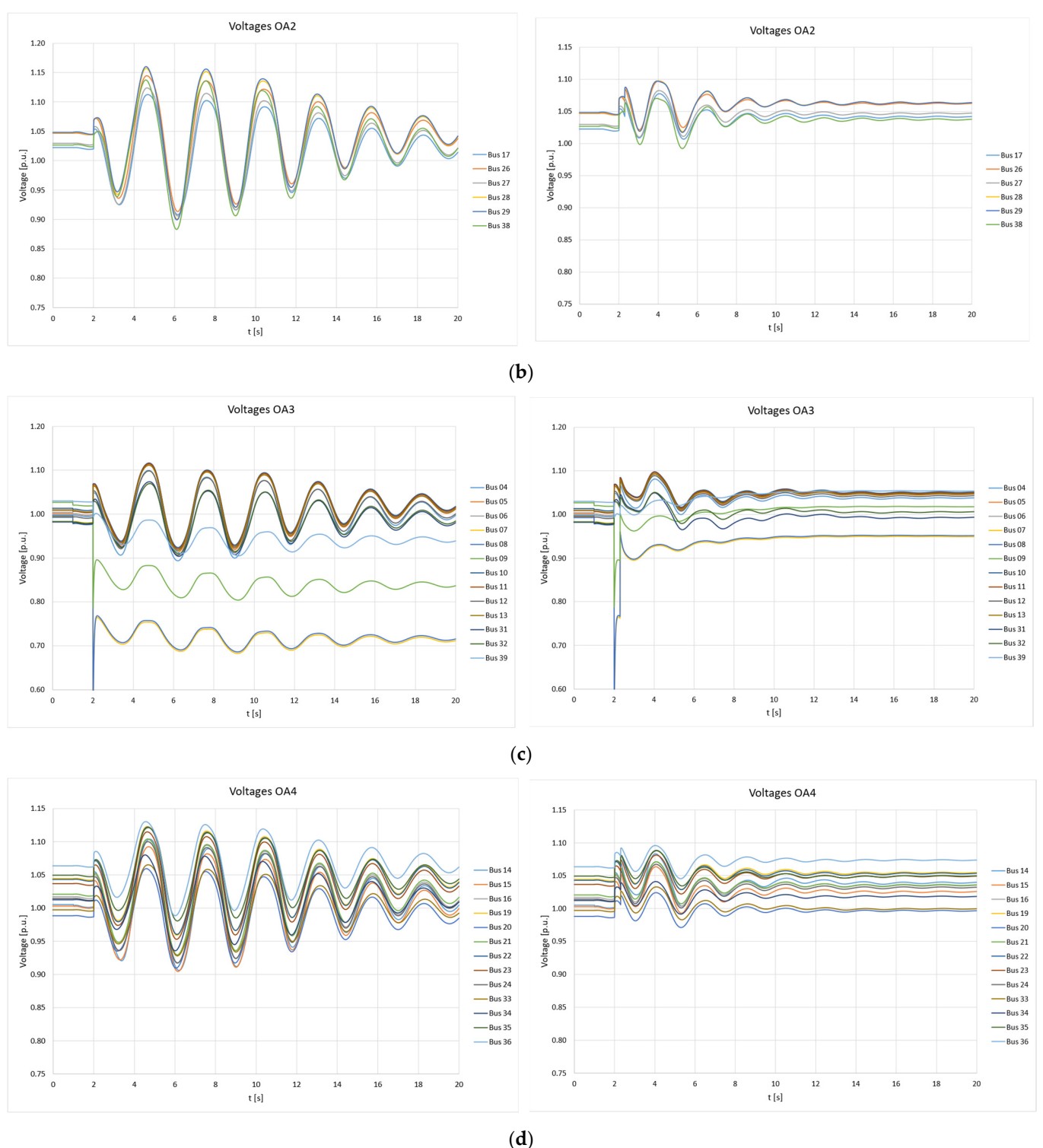

**Figure 20.** Voltage results without and with control actions (outages of lines 5–8 and 6–7). (**a**) Voltages OA1. (**b**) Voltages OA2. (**c**) Voltages OA3. (**d**) Voltages OA4.

## 8. Conclusions

In this paper, a novel method for voltage stability based on P–V curves and analysis of angular indexes calculated by synchrophasors was addressed.

It was observed that the center of angle (COA) and delta of center of angle (DCOA) yield the characterization of an electrical power system. It was even possible to determine

if a contingency is critical when a detailed prior characterization of the system is available. In addition, the COA proved to be an important index to determine the location of the occurrence of the events, and where control actions should be carried out to remedy the stress problem in this case.

A limitation of the stationary analysis is the non-convergence against critical events, which would not yield the characterization of the angular index. Based on this limitation, the stationary analysis does not provide the necessary information for the contingencies of rare occurrences and the high impact in which convergence is not achieved.

**Author Contributions:** The authors contributed to the paper in the following capacities: conceptualization, G.J.L.; methodology, J.W.G.; analysis, I.A.I. and O.H.V.; simulation, H.A.C. All authors have read and agreed to the published version of the manuscript.

**Funding:** This research was funded by the program: "Estrategia de transformación del sector energético colombiano en el horizonte de 2030" funded by the research call 778 of MinCiencias Ecosistema Científico. Contract FP44842-210-2018.

**Data Availability Statement:** The data presented in this study are available on request from the corresponding author. The data are not publicly available due to the data use agreement.

**Acknowledgments:** The authors would like to thank the support to Carlos Eduardo Restrepo, Andrés Felipe Eusse and Camilo Andrés Villarreal Rueda.

**Conflicts of Interest:** The authors declare no conflict of interest.

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
