# Peer review of "Voltage Stability Control Based on Angular Indexes from Stationary Analysis"

_energies, doi:10.3390/en15197255_

Round 1

Reviewer 1 Report

This paper presents a novel methodology for the calculation of angular indexes of an electrical system from stationary analysis, using load flow and nose curves (P-V) in each of the buses of the system to perform control actions and preserve or improve voltage stability. Overall paper is well organized and well presented. However, the introduction section needs to be improved by comparing more state-of-the-art research on this topic. The diagrams of this paper need to be improved. 

Author Response

Best regards

Reviewer 2 Report

The article proposes a novel methodology for the calculation of angular indexes of an electrical system from stationary analysis, using load flow and nose curves (P-V) in each of the buses of the system to perform control actions and preserve or improve voltage stability. So the topic is actual and adequate to scientific journal (in this case Energies). The methods are presented in clear way. Results are sufficient. 

But here are some recommendations to improve the paper:

-> In the abstract, the results should be quantified also. 

-> the number of papers from 2022 is not sufficient (only one at present form). Please extend with at least 3 to ensure that the literature review is present.

-> Results should be discussed (as a separate section) in the context of secondary literature

-> please provide adequate references for equations given in all sections.

Author Response

Best regards

Reviewer 3 Report

The authors propose a novel methodology for the calculation of angular indexes of an electrical system from stationary analysis, using load flow and nose curves (P-V) in each of the buses of the system to perform control actions and preserve or improve voltage stability.

Questions:

1. Authors need to do a better bibliographic review. Most of the cited articles are more than 6 years old and recently many methods for estimating and improving the voltage stability margin of power systems have been proposed.

2. Is the proposed method limited to power systems with synchronous generation only? Because the method depends on angle measurements and inertia constant of synchronous generators. Currently, there is a high penetration of wind and solar energy sources in power systems around the world, the method proposed by the authors does not seem to meet the current operating conditions of modern power systems.

3. Different directions of load growth result in different PV curves and consequently different voltage stability margin values. Apparently the authors considered a single direction of load growth in the static analysis and evaluated the proposed method in real time. However, in real-time the power system can take different directions of load growth and thus analyzes conducted in static (offline) analysis may not be suitable for real-time operation.

4. Did the authors consider the reactive power limits of synchronous generators? Because this reactive power limitation affects the tracing of the PV curves and may decrease the voltage stability margin values if this limitation was not considered.

5. The authors do not present comparative results with existing methods in the literature to assess the benefits and weaknesses of the proposed method.

Author Response

Best regards

Round 2

Reviewer 2 Report

Paper has been well revised. I recommend to publish it .

Reviewer 3 Report

The authors propose a novel methodology for the calculation of angular indexes of an electrical system from stationary analysis, using load flow and nose curves (P-V) in each of the buses of the system to perform control actions and preserve or improve voltage stability.

The article has been improved, the contribution is good and all questions have been effectively answered.